# Research Progress on Multifunctional Modified Separator for Lithium–Sulfur Batteries

**DOI:** 10.3390/polym15040993

**Published:** 2023-02-16

**Authors:** Ying Wang, Rui Ai, Fei Wang, Xiuqiong Hu, Yuejing Zeng, Jiyue Hou, Jinbao Zhao, Yingjie Zhang, Yiyong Zhang, Xue Li

**Affiliations:** 1National and Local Joint Engineering Laboratory for Lithium-Ion Batteries and Materials Preparation Technology, Faculty of Metallurgical and Energy Engineering, Kunming University of Science and Technology, Kunming 650093, China; 2College of Electrical Information Engineering, Panzhihua University, Panzhihua 617000, China; 3College of Vanadium and Titanium, Panzhihua University, Panzhihua 617000, China; 4Collaborative Innovation Center of Chemistry for Energy Materials, Provincial Government New Energy Vehicle Power Technology Joint Engineering Laboratory, State Key Laboratory of Physical Chemistry of Solid Surfaces, School of Chemistry and Chemical Engineering, Xiamen University, Xiamen 361005, China

**Keywords:** lithium–sulfur battery, modified separator polysulfides

## Abstract

Lithium–sulfur batteries (LSBs) are recognized as one of the second-generation electrochemical energy storage systems with the most potential due to their high theoretical specific capacity of the sulfur cathode (1675 mAhg^−1^), abundant elemental sulfur energy storage, low price, and green friendliness. However, the shuttle effect of polysulfides results in the passivation of the lithium metal anode, resulting in a decrease in battery capacity, Coulombic efficiency, and cycle stability, which seriously restricts the commercialization of LSBs. Starting from the separator layer before the positive sulfur cathode and lithium metal anode, introducing a barrier layer for the shuttle of polysulfides is considered an extremely effective research strategy. These research strategies are effective in alleviating the shuttle of polysulfide ions, improving the utilization of active materials, enhancing the battery cycle stability, and prolonging the cycle life. This paper reviews the research progress of the separator functionalization in LSBs in recent years and the research trend of separator functionalization in the future is predicted.

## 1. Introduction

Since the beginning of the 21st century, the frequency of natural disasters, the number of people, and economic losses have increased exponentially compared with 1980–1999 [1]. Human beings continue to develop and use fossil energy. However, there is “no doubt” that the carbon dioxide and other harmful gases produced by the burning of fossil fuels contribute to global warming [2]. Against this background, the climate change targets set by the Paris Agreement have achieved global consensus. In June 2022, more than 130 countries around the world have announced carbon neutrality goals or plans. As the primary “contributor” to global greenhouse gas emissions, the energy sector has become an important reform area for countries around the world to promote carbon neutrality [3]. At present, many countries have promoted and used renewable and clean energy, such as solar energy, wind energy, tidal energy, geothermal energy, etc. However, these new energies are often discontinuous and geographically restricted, so it is difficult to directly add them to the grid for use. Therefore, it is crucial to accelerate the promotion of large-scale, long-term energy storage with a low cost and high safety performance.

LSBs are recognized as one of the second-generation electrochemical energy storage systems with the most potential due to the low cost, high reserves, environmental friendliness, and high energy density of its active material sulfur, and they have attracted the attention of many researchers [4,5,6]. The theoretical specific capacity of sulfur is 1675 mAh g^−1^ (based on S_0_-S^2−^calculation), and, combined with the average working voltage of the LBS (2.15 V vs. Li^+^/Li) and the theoretical capacity of the lithium anode (based on the 3860 mAh g^−1^ at the Li^+^-Li_0_ calculation), it can be calculated that the theoretical energy density of the LBSs is as high as −2500 Wh kg^−1^, which is about an order of magnitude higher than that of traditional lithium batteries [7,8,9,10]. Sulfur in the earth’s crust is abundant and its price is low (as low as 1000 yuan per ton of sulfur); at the same time, LSBs are a kind of “green battery”, and, regarding both the sulfur electrode material itself and the process of charging and discharging, almost no harmful substances to the environment are produced [11,12]. Therefore, Lithium-sulfur battery is considered as one of the most attractive secondary battery system.

## 2. Reaction Mechanism of LSBs

The structure of LSBs is the same as that of lithium-ion batteries, and is mainly composed of the sulfur cathode, separator, lithium metal anode, and organic electrolyte. The operating principle of LSBs is different from that of traditional lithium–ion batteries [13,14,15]. During the charging and discharging process, the sulfur cathode has a complex and multistep redox reaction, which is also accompanied by a complex phase transition process of lithium polysulfides.

When the LSBs are charged and discharged, the electrochemical redox reaction that occurs at the sulfur cathode and the lithium metal anode is abbreviated as follows:Sulfur cathode: S + 2Li^+^ + 2e^−^ ↔ Li_2_S(1)
Lithium metal anode: 2Li ↔ 2Li^+^ + 2e^−^(2)
Total reaction equation: S + 2Li ↔ Li_2_S(3)

This is a two-electron reaction. The theoretical specific capacity of the sulfur is 1675 mAh g^−1^, and the theoretical energy density of LSBs is as high as −2500 Wh kg^−1^ (as shown in Figure 1a), which is 5–7 times that of current commercial lithium-ion batteries [16]. However, this electrochemical reaction does not take into account the effects of the dissolution of lithium polysulfide intermediates produced during the cycle of lithium–sulfur batteries. The electrochemical reaction of sulfur cathode is a complex process in the lithium-sulfur battery of ether-based organic electrolyte system, which will produce lithium polysulfide intermediates with different chain lengths and can be dissolved in organic electrolyte (Li_2_S_n_, n > 2). A series of side reactions occur when the dissolved lithium polysulfide intermediate body shuttles to the negative electrode, resulting in the loss and redistribution of the active material, rapid attenuation of battery capacity, and serious self-discharge.

During discharging, the solid phase elemental S_8_(S) is gradually reduced according to Equations (4)–(9), and the reduced intermediate S_n_^2−^(8 ≥ n ≥ 4) combines with lithium ions to form a long chain of Li_2_S_n_, which is easily soluble in the organic electrolyte and diffuses into the electrolyte due to the concentration gradient. With the progression of discharge, the long-chain S_n_^2−^ (8 ≥ n ≥ 4) is further reduced to short-chain polysulfides S_2_^2−^ and S^2−^, which bind to lithium ions to form Li_2_S_2_ and Li_2_S that are almost insoluble in the electrolyte, such as in Equations (10) and (11).
S_8(S)_ ↔ S_8(L)_(4)
1/2S_8(I)_ + e ↔ 1/2S_8_^2−^(5)
(6)$$3/2S_{8}2^{-}+e\;↔\;2S_{6}2^{-}$$
S_6_^2−^ + e ↔ 3/2S_4_^2−^(7)
1/2S_4_^2−^ +e ↔ S_2_^2−^(8)
1/2S_2_^2−^ +e ↔ S^2−^(9)
S_2_^2−^ + 2Li^+^ ↔ Li_2_S_2_(10)
S^2−^ + 2Li^+^ ↔ Li_2_S(11)

The charge–discharge curve of LSBs is shown in Figure 1 [17,18], and the corresponding specific electrochemical reactions are as follows. As shown in Figure 1b, the general discharge process can be mainly divided into four parts: the high-voltage discharge platform, first slash area, low-voltage discharge platform, and final slash area. First, the high-voltage discharge platform region (−2.3 V) and the first slash area mainly correspond to the sulfur cathode electrochemical reaction (5)–(7); S_8_ molecules bind electrons and lithium ions moving from the negative electrode to transform into Li_2_S_8_, Li_2_S_8_ with a ring opening mechanism, and are soluble in organic electrolytes, promoting the utilization and reaction kinetics of internal S. Subsequently, as the discharge continues, long-chain polysulfides continue to generate, and its concentration increases, resulting in an increased viscosity in the electrolyte. When the viscosity increases to a certain value, the lithium-ion moving becomes more and more difficult, resulting in concentration polarization, voltage drop, and charge, and the discharge curve enters into the first slash area; at this time, the polysulfide intermediates are dominated by Li_2_S_4_. As the discharge continues, the discharge curve enters the low-voltage platform area (−2.1 V). The long-chain polysulfides slowly transition to the short-chain polysulfides (S_2_^2−^ and S^2−^), and, finally, combine with the lithium ions to form insoluble Li_2_S_2_ and Li_2_S. At this time, as the impedance increases and the voltage decreases, the discharge curve enters the final slash region, and the discharge terminates, corresponding to the sulfur cathode electrochemical reaction (8)–(11).

At the same time, as shown in Figure 1b, the discharge process can also be divided into four reduction zones according to the changes in the phase composition of sulfur species: Region (I) is the solid–liquid two-phase reduction zone from S_8_(s) to Li_2_S_8_ (corresponding to reactions (4) and (5)), which shows a high-voltage discharge platform at 2.2–2.3 V. In this region, the formed Li_2_S_8_ is dissolved in the liquid electrolyte and becomes a liquid-phase active substance. This process leaves many voids at the positive electrode. Region (II) is a single-phase reduction region reduced from liquid-phase Li_2_S_8_ to short-chain lithium polysulfides (corresponding to reactions (6) and (7)). In this region, the voltage of the battery drops sharply. With the decrease in the length of the lithium polysulfide chain and the increase in the number (concentration) of polysulfides, the viscosity of the solution increases gradually. At the end of this discharge region, the viscosity of the solution reaches the maximum. Region (III) is the liquid–solid two-phase reduction region from dissolved short-chain lithium polysulfide to insoluble Li_2_S_2_ or Li_2_S (corresponding to reactions (8)–(11)). This area shows a low-voltage discharge platform of 1.9–2.1 v, which is the main contribution area of the lithium–sulfur battery capacity. Region (IV) is the solid–solid single-phase reduction region from insoluble Li_2_S_2_ solid to Li_2_S. Due to the non-conductivity and insolubility of Li_2_S_2_ and Li_2_S, the process kinetics are slow and the degree of polarization is high. In the above four regions, the redox shuttle in regions (I) and (II) is the most significant, during which the battery self-discharge rate is high and the battery capacity fades largely. Region (III) is the main contribution region of the lithium–sulfur battery capacity. When the generation of Li_2_S is dominant in region (III), region (IV) becomes very short or even disappears. In fact, during the discharge of LSBs, the conversion process of sulfur active substances is not carried out step by step in strict accordance with the above Equations (4)–(11), and the actual reaction is more complex, determined by the characteristics of polysulfide itself. In addition to the electrochemical redox reaction, a chemical disproportionation reaction also occurs in the discharge process of polysulfides.

In 1962, Danyta and Jukiusz [19] proposed to use sulfur as the cathode material of dry batteries, and LSBs began to enter the field of vision of scientific researchers. However, due to the complexity of the electrochemical principle of lithium–sulfur batteries and some serious problems in the anode and cathode itself, the performance improvement and commercial application of lithium–sulfur batteries have been seriously hindered, so they have not been widely considered. The challenges of lithium–sulfur battery research will be elaborated as follows.

## 3. Challenges of Lithium Batteries

Due to the low cost and environmental friendliness of the active material S, as well as its high weight/volume energy density and low operating voltage, lithium–sulfur batteries can be used in electric vehicles on a large scale. In addition, the batteries’ life in electronic devices, such as smartphones or laptops, can be further extended. However, in the actual commercialization process of lithium–sulfur batteries, the sulfur cathode, lithium metal anode, electrolyte, and ecological and economic aspects of the preparation process have encountered various problems and challenges.

### 3.1. Poor Electrical Conductivity of the Active Material

As an insulator, at 20 °C, the resistivity of lithium–sulfur battery cathode active material S reaches as high as 2 × 10^23^ μm or its conductivity is as low as 5 × 10^−30^ S cm^−l^. The final discharge product, lithium sulfide (Li_2_S), is also an insulator and has poor electrical conductivity [20]. Poor conductivity will lead to a low utilization rate of the active material, as the actual specific capacity is not high. In order to ensure the rapid conduction of electrons and lithium ions, a large number of conductive additives must be added during the preparation of the sulfur cathode. However, the conductive additive is an inactive substance and cannot provide an effective capacity, resulting in a reduction in the overall capacity of the cathode material. At the same time, the addition of a large number of conductive additives will also increase the difficulty of preparing a uniformly dispersed cathode material slurry.

### 3.2. The Polysulfides Shuttle

In particular, the lithium polysulfide produced in the charge and discharge process of lithium–sulfur batteries is highly soluble in ether electrolyte. Due to the electric field force and concentration gradient, the polysulfide ions dissolved in the organic electrolyte can freely move back and forth between the cathode and anode, and this migration behavior is called the “shuttle effect”. The shuttle effect of polysulfide ions is particularly obvious in the area of the high-voltage charging platform, because the long-chain lithium polysulfide can migrate through the separator to the anode side, where the lithium metal anode is converted into short-chain lithium polysulfide by electrochemical and chemical reduction (Equations (12) and (13)). Short-chain lithium polysulfide can be reverse-diffused to the cathode side and be reoxidized long-chain lithium polysulfide. It goes on and on and on, sometimes exceeding the theoretical capacity of sulfur. The polysulfide lithium through the separator reacts with the lithium metal as shown in Equations (14) and (15), forming Li_2_S and Li_2_S_2_ passivation layers at the negative electrode, which, on the one hand, consumes the active substances at the cathode, and, on the other hand, leads to the corrosion and passivation of the anode and also reduces the coulomb efficiency of the battery.
(n − 1) Li_2_S_n_ + 2Li^+^ + 2e^−^ ↔ n Li_2_S_n−1_(12)
(n − 1) Li_2_S_n_ + 2Li ↔ n Li_2_S_n−1_(13)
2Li + Li_2_S_n_ ↔ Li_2_S↓ + Li_2_S_n−1_(14)
2Li + Li_2_S_n_ ↔ Li_2_S_2_↓ + Li_2_S_n−1_(15)

### 3.3. Volume Expansion

As shown by reaction (1), the final product of the sulfur cathode during the charge–discharge cycle is S and Li_2_S, and there are significant differences in density between these two substances. Among them, the density of S is 2.07 g cm^−3^, while the density of Li_2_S is 1.66 g cm^−3^. When being converted into volume, the sulfur cathode undergoes an about 80% volume change during the cycle. Due to the huge volume expansion, the electrode will crack and form cracks, resulting in the shedding of the active material of the electrode or the destruction of the overall conductive network of the electrode, which eventually leads to the deterioration of the battery performance or even failure. To buffer the volume expansion, flexible carbon composites or composite materials partially filled with S and binders with stronger toughness can be used.

### 3.4. Corrosion and Dendrite Growth of Lithium Anode

As the anode of the lithium sulfur battery, lithium metal has a strong chemical reactivity and reacts with almost all substances. Repeated reactions occur after contact with lithium polysulfide in the electrolyte, lithium metal corrosion and powder, reducing the utilization rate of the anode. In addition, the heterogeneous deposition of lithium ions often leads to the growth of lithium dendrites during the nucleation and growth of lithium. Lithium dendrite penetrates the diaphragm and directly contacts the anode and cathode, resulting in short circuit and affecting the safety of the battery.

The “Shuttle effect” is one of the key points and difficulties in the research and commercial application of lithium–sulfur batteries. In the past, the majority of researchers liked to start from the cathode material research, by optimizing and improving the cathode to slow down the shuttle of polysulfide ions. Some researchers also use lithium anode protection and other methods to slow the shuttle of polysulfide ions. However, it is difficult to obtain an efficient, high-yield, economical, and safe commercial solution either to optimize cathode materials or to adopt lithium anode protection. In recent years, many researchers have tried to effectively inhibit the repeated shuttle of lithium polysulfide from the aspect of separator modification and achieved certain results. The following will elaborate systematically the research from domestic and foreign researchers on the separator of lithium–sulfur batteries in recent years.

## 4. Research Progress in the Engineering Design and Function of Lithium–sulfur Battery Separators Modification

As shown in Figure 2, the separator located between the anode and cathode is a key component of lithium–sulfur batteries. It can provide a channel for lithium-ion transmission and prevent contact short circuit between anode and cathode electron transmission. In the lithium–sulfur battery system, the separator is the only way for soluble polysulfide ions to enter the lithium anode region. However, ordinary commercial polymer separators are not effective at preventing the transport of polysulfide ions in the electrolyte, resulting in a “shuttle effect”. Therefore, starting from the separator between the cathode and anode of the lithium–sulfur battery, under the premise of ensuring the insulation of the lithium-ion channel and electron, the separator is given the ability to block the passage of polysulfide ions, and the dissolved polysulfide lithium is restricted in the cathode region, to achieve the purpose of slowing down the shuttle effect and protecting the lithium metal anode. According to literature reports, there have been some research and utilization of functional diaphragms in lithium–sulfur batteries, mainly focusing on the following aspects:

On the one hand, the separator is modified by electrostatic reaction (sulfonic acid, carboxylic acid and BaTiO_3_ in Nafion), steric hindrance and ion selection. The modified separator can effectively inhibit the diffusion of lithium polysulfide and improve the coulomb efficiency. On the other hand, materials with structural design and polar materials [21,22,23,24] are used to modify the separator, and the diffusion of lithium polysulfide is inhibited by physical adsorption [25,26,27] and chemical adsorption (${\mathrm{Ti}}_{3}C_{2}$ and oxygen/nitrogen functional groups) of the microporous structure [28] and improvements in the conductivity of the cathode materials. At the same time, the electronic conduction of the contact surface with the positive electrode is realized to reduce the interface impedance of the positive electrode side, achieving the secondary utilization of lithium sulfide and improve the cycle stability and active substance utilization rate of the Li-S battery.

With the gradual deepening of the research on lithium–sulfur batteries, the combination of a variety of strategies and the synergy of a variety of functions to design a multifunctional separator, can have a more significant effect on the shuttle effect of inhibition and lithium–sulfur battery performance [29,30,31]. At present, according to literature reports, studies on the separator between the anode and cathode mainly focus on the following aspects.

### 4.1. Improvement of the Adsorption Capacity of Polysulfides of Functionalized Separators

Initially, the researchers modified the conductivity of the separator to prevent polysulfide from traveling to the lithium metal anode. This kind of inhibition of the polysulfide shuttle through the adsorption of polysulfide ion by the conductive layer has attracted wide attention. Subsequently, the majority of scientific research workers in the premise of ensuring the conductivity of the separator have put forward a variety of schemes to modify the separator to improve the adsorption capacity of lithium polysulfide, so as to achieve a better sulfur-fixation effect. In recent years, researchers have conducted a lot of scientific research to improve the adsorption capacity of separator materials. There are three methods to improve the adsorption performance of the conductive layer:(1)The abundant microporous and mesoporous structures can provide more sites for the electrochemical conversion of polysulfides, improve the efficiency of electrochemical reactions, and play a role in physical sulfur fixation. Polar inorganic compounds have an excellent adsorption capacity for polysulfides, so the introduction of polar materials into the conductive modification layer can more effectively adsorb dissolved polysulfides and play an efficient sulfur fixing effect. However, the conductivity of most inorganic compounds is very poor, and increasing the doping amount of polar materials is bound to reduce the conductivity of the modified layer and sacrifice the electrochemical activity of the modified layer. Therefore, scientific researchers try to achieve the conductive functionalization of polar materials or select polar adsorption materials with excellent conductivity to modify the separator to improve the comprehensive performance of lithium–sulfur batteries. Pengyu Li et al. [32] first reported on the separator (HGSCS) of hollow graphene ball-coated lithium–sulfur batteries. Hollow graphene balls (HGSs) have the effect of adsorbing and fixing lithium polysulfide, effectively blocking lithium polysulfide from shuttling back and forth to the cathode and anode, greatly reducing the shuttle effect. The hollow graphene ball modified lithium–sulfur battery separator exhibits excellent electrochemical properties, discharging at 0.2 times, and its initial specific capacity is as high as 1172.3 mAh g^−1^, the battery capacity remains at 824.1% after 200 cycles, and the capacity retention rate is as high as 94.41%.(2)Polar inorganic compounds have an excellent adsorption capacity for polysulfides, so the introduction of polar materials into the conductive modification layer can more effectively adsorb dissolved polysulfides and have an efficient sulfur solidification effect [33]. In the literature, Al_2_O_3_ [34,35], SiO_2_ [36,37], MnO_2_ [38], V_2_O_5_ [39], Mg_0_._6_Ni_0_._4_O [40], hydroxyapatite, Fe_2_O_3_ [41], NbN [42], TiO_2_ [43], MoO_3_ [44], and so on are used to enhance the adsorption and sulfur fixation performance of the conductive layer. Adding polar inorganic compounds can greatly improve the sulfur fixing effect of the modified layer. However, most of the inorganic compounds have poor conductivity and the increase of the doping amount of polar materials is bound to reduce the conductivity of the modified layer and sacrifice the electrochemical activity of the modified layer. Therefore, researchers try to achieve the conductive functionalization of polar materials or select polar adsorption materials with excellent conductivity to modify the separator to improve the comprehensive performance of lithium–sulfur batteries. For example, Yang Wang et al.(Figure 3A,B) [45] successfully modified the polypropylene (PP) separator for the first time by using the binary transition metal oxide MnFe_2_O_4_ prepared by co-precipitation. The bifunctional modified separator decorated with MnFe_2_O_4_ and conductive acetylene black (AB) effectively hinders the diffusion of polysulfide compounds due to the synergy of chemical adsorption and physical limits. The sulfur load was about 0.7–1 mg/cm^2^.The battery using the modified separator has a magnification capacity of 920 mAh g^−1^ and 845 mAh g^−1^ under 1 C and 2 C conditions, respectively, the capacity is divided into 625 mAh g^−1^ after cycling 500 laps under 1 C conditions, and the capacity attenuation rate per revolution is 0.074%. Inigo Garbayo et al.(Figure 3C) [46] proposed a way to solve the polysulfide shuttle, that is, the use of alumina nanocoating in the electrode–electrolyte interface, thickness in the 10 nm range, and growth by magnetron sputtering, which can be used as an impervious layer of polysulfides between the electrode and the electrode, a solid polymer electrolyte separator. Through constant current cycle experiments, it is proved that the coating as a polysulfide barrier layer has the effect of reducing the polysulfide shuttle and prolonging the cycle life of the battery. Maryam Sadat Kiai et al. (Figure 3D) [47] use a novel liquid graphene oxide (L-GO) binder instead of the standard polyvinylidene fluoride (PVDF) binder to deposit silica (SiO_2_), titanium dioxide (TiO_2_), and poly1,5-diaminoanthraquinone (PDAAQ) on the glass fiber separator to significantly delay the growth of lithium dendrite by forming chemical bonding, alleviating the shuttle effect. Assembling the battery using SiO_2_/L-GO, TiO_2_/L-GO, and PDAAQ/L-GO modified separators, the capacities remain at 1020, 1070, and 1190 mAh g^−1^ after 100 cycles, respectively. The mass loading of sulfur in the cathode was ~2.5 mg cm^−2^.

(3)The heteroatomic doping of carbon materials, by changing the charge distribution state on the surface of carbon materials, can provide the polarity site of adsorption of polysulfides and improve the sulfur-fixation performance of the conductive barrier layer [48,49]. In the literature publicized, the use of B [50], N, O [51], P [52], S [53,54], and other heteroatoms to dope carbon materials and the energy density, cycle life and magnification performance of lithium–sulfur batteries have a significant effect. Xiong Song (Figure 4) [55] et al. introduced a new lightweight multifunctional layer modified commercial polypropylene (PP) separator composed of one-dimensional porous iron embedded in nitrogen-doped carbon nanofibers (Fe-N-C) and two-dimensional graphene sheets. This new modified Fe-N-C/G@PP separator has four main advantages: (i) due to its unique porous intercalation structure and highly improved wettability, the Fe-N-C/G integrated layer can maintain a high transfer rate of lithium ions; (ii) the highly conductive Fe-N-C nanofibers can provide a strong LiPS chemical fixation; (iii) The Fe-N-C/G modified layer can be used as a three-dimensional conductive backbone to promote the reuse of sulfur species captured in the Fe-N-C/G layer; (iv) the lightweight Fe-N-C/G layer (only 0.083 mg cm^−2^) does not affect the energy density of Li-S batteries. The results show that the electrochemical properties of Li-S batteries using an Fe-N-C/G@PP separator are significantly improved, showing a good cycle performance and a high reversible capacity retention rate. The mass loading of the Fe-N-C/G modified layer onto the PP separator is 0.083 mg cm^−2^. After 500 cycles of charge and discharge, the battery capacity remains at 601.9 mAh g^−1^, and the capacity decay per revolution is only 0.053%. Through the study, it was found that diatomic doping has a synergistic improvement on the adsorption capacity of conductive carbon materials to lithium polysulfide.

In addition, other researchers apply graphene to the field of lithium–sulfur batteries and use its excellent conductivity and large surface-to-diameter ratio to solve the conductivity of the cathode material, the diffusion of the positive pole polysulfides out of the positive electrode, and the preparation of self-supporting cathode materials [56,57,58]. The use of the graphene-modified separator, while giving the modified layer ultra-high conductivity, physically hinders the shuttle of polysulfides through the space steric resistance effect, and then, by doping the heteroatoms of graphene or introducing other functional materials, the comprehensive performance of lithium–sulfur batteries can be greatly improved. Liuli Zeng et al.(Figure 5A–D) [59] prepared a graphene oxide and ferrocene co-modified polypropylene (PP) separator. Graphene oxide on the functionalized separator can physically adsorb polysulfides, while ferrocene components can effectively promote the conversion of adsorbed polysulfides. Thanks to the combination of these beneficial features, the separator exhibits excellent battery performance, achieving a high reversible capacity of 409 mAh g^−1^ after 500 cycles at 0.2 C magnification (The areal sulfur loading on the electrode was ~1.3 mg cm^−2^). Since nitrogen-doped graphene (NG) has “Lithium-loving” properties and VSe2 ultrafine nanocrystals have excellent “sulfur-loving properties”, Wenzhi Tian et al. (Figure 5E) [60] fixed VSe_2_ on nitrogen-doped graphene to modify the battery separator to alleviate the polysulfide shuttle. VSe_2_ nanocrystals provide a large number of active sites for the chemical adsorption of polysulfides and are conducive to the nucleation and growth of Li_2_S. At the same time, the two materials are used as the barrier layer of the polysulfide shuttle, and the lithium polysulfide is co-adsorbed, showing a better sulfur-fixation effect. Through the modification of the separator, use of synergistic sulfur fixation, excellent conductivity, and physical barrier, the electrochemical properties are greatly improved, with an excellent long cycle stability and ultra-high magnification capacity of up to 8 C. In addition, when the sulfur load is as high as 6.1 mg·cm^−2^, the regional capacity reaches 4.04 mAh cm^−2^.

Graphene materials have significant advantages in the performance improvement of lithium–sulfur batteries, and many researchers have used graphene materials to carry out a large number of research works on the functionalization of separators. Various heteroatomic doping [61,62,63], inorganic compound doping [64], polymer coating [65], and other strategies have obtained excellent performances of lithium–sulfur batteries, fully proving that graphene materials play an important role in inhibiting the shuttle of polysulfides and improving the electrochemical activity of lithium–sulfur batteries.

### 4.2. Multifunctional Separator with Catalytic Function

All kinds of adsorbent materials reported in a large number of studies can effectively absorb polysulfide ions, which plays a crucial role in improving battery performance. However, to ensure sufficiently high energy density and conductivity, the addition of polar adsorbent materials must be limited. Therefore, the amount of polysulfide ions adsorbed by the modified layer is limited. In the charging and discharging process of the lithium–sulfur battery, the kinetic process of polysulfide ion conversion is slower than the irreversible thermodynamic process, which leads to the uneven deposition and agglomeration of lithium sulfide, which affects the discharge capacity and cycle life of the battery. Therefore, increasing the conversion rate between long-chain polysulfide lithium and lithium sulfide in the charging and discharging process can fundamentally reduce its dissolution shuttle in the electrolyte and exert higher discharge capacity [66]. On this basis, catalytic materials that can promote the redox reaction of polysulfides are used to improve the conductive barrier layer [67,68]. This kind of multifunctional separator has three advantages: (1) improving the electron conductivity and the utilization rate of active substances; (2) reducing the dissolution and diffusion of polysulfides and effectively inhibiting the shuttle effect; (3) accelerating the reaction kinetics of polysulfides.

Li Xuting(Figure 6) [69] et al. synthesized modified separator using FeTaPc self-assembly process adsorbed on RGO. The modified separator can effectively capture cathode polysulfides and catalyze sulfur oxidation–reduction conversion with a high lithium-ion transfer number, inhibiting dendrite formation and lithium anode surface corrosion. Highly catalytic FeTaPc and LiPSs molecules exhibit the geometry of amphiphilic sulfur/lithium adsorption, which promotes lithium-ion transport, significantly improving the number of lithium-ion transfers, and effectively accelerates the kinetic process of polysulfides conversion. The lithium–sulfur battery was assembled with a sulfur loading of 1.5 mg cm^−2^. The initial discharge specific capacity reached 1203 mAh g^−1^ at a 1 C rate, and the battery capacity remained at 862 mA g^−1^ after 500 cycles. The average capacity decay rate per cycle is 0.056%, the Coulombic efficiency remains above 96%, and the battery stability performance is good (Figure 6). Lina Jin et al. [70] used ZIF8 as the core and ZIF67 as the shell to prepare Co-N doped hollow carbon nanocages for the modification of a lithium–sulfur battery separator. On the one hand, the active site doping of metal Co and N in the coating can increase the chemical catalytic effect based on the physical adsorption of polysulfide. On the other hand, the conductive network of carbon nanotubes integrated on the surface of hollow carbon nanocages can improve ionic conductivity. Co-n-c nanocages accelerate the REDOX kinetics of polysulfides. Even after 200 cycles at a sulfur load of 3 mg cm^−2^, 0.5 C, the battery still shows a high reversible capacity of 725 mAh g^−1^ with a capacity retention rate of 88%. Especially when the sulfur load is increased to 4.7 mg cm^−2^, the battery can still cycle stably more than 400 times at 1 C.

### 4.3. Functional Separator with Electrostatic Repulsion Effect

Using the simple electrostatic repulsion principle of the mutual repulsion of the same charge, the use of negatively charged groups can effectively shield the shuttle of polysulfide anions. In addition, the separator with a negative group can selectively pass through Li^+^, which improves the lithium-ion conductivity and lithium-ion migration number. The studies about this part publicized in the literature are mainly divided into two categories:(1)Sulfonated polystyrene (PSS) with ultra-high sulfonic acid group content [71] is used for the functionalization of polyolefin separators. Haritha Hareendrakrishnakumar et al.(Figure 7A,B) [72] proposed a lithiated PEDOT: PSS-coated modified separator that has a dual role including inhibiting polysulfide shuttles and promoting Li+ selective diffusion. The negatively charged sulfonic acid group in PSS has an electrostatic shielding effect on the soluble high-order polysulfides through Coulomb repulsion, while the strong negatively charged atoms (O and S) in PEDOT form chelate coordination structures with insoluble lithium sulfides. The Li+ -PEDOT: PSS@CG separator (lithiated PEDOT: PSS-coated Celgard) has good electrolyte wettability, ionic conductivity, and interfacial properties. The battery has a sulfur load of 4.1 mg cm^−2^ and a discharge rate of 0.5 C, with an initial discharge capacity of 1096 mAh g^−1^ and a discharge capacity of 911 mAh g^−1^ after 500 cycles. Chunyang Zhou et al. (Figure 7C,D) [73] prepared a C-PVA/PAA-Li composite nanofiber separator by electrostatic spinning, thermal crosslinking, and an in situ lithium process. The C-PVA/PAA-Li composite nanofiber separator has a good porous structure and high ionic conductivity. It can prevent the polysulfide anion shuttle through electrostatic repulsion, reduce the charge transfer resistance, accelerate the migration of Li+, and inhibit the growth of lithium dendrites. The battery has a sulfur load of 1.5–2.0 mg cm^−2^. After 400 cycles of 0.2 c, the decay rate of the battery is only 0.08%, and the capacity of the battery is still 633 mAh g^−1^ at the current density of 3 C.

(2)Because of the excellent performance of the conductive coating between positive separators, some researchers also try to combine the improvement of conductive performance with the electrostatic shielding strategy: on the one hand, the shuttling and deposition of polysulfides can be reduced by electrostatic shielding; on the other hand, it can provide a place for the electrochemical reaction of polysulfides, activating the deposited active substances, and also greatly help to reduce the interface resistance. Chengwei Songet al.(Figure 8) [74] proposed a reasonable design scheme for integrating conductive MWCNTs multilayers and PEI @ MWCNTs-CB layers on conventional PP separators (PEI@MWCNTs-CB/MWCNTs/PP briefly denoted as PMS) to improve the performance of Li-S batteries. The PEI @ MWCNTs-CB layer on the surface reduces the shuttle and deposition of polysulfides through the electrostatic shielding effect, thus stabilizing the cycle performance, while the internal conductive carbon layer is conducive to providing the electrochemical reaction sites of polysulfides in the separator and activating the deposited active substances. Due to the synergistic effect of dual-structure and multifunctional PMS modified separators, it shows high reversible capacity and superior long cycle performance. Especially under the condition of a high sulfur load (sulfur content reaches 80%), when discharging at 1 A·g^−1^ current density, the cycle exceeds 120 cycles, and the capacity of each cycle is reduced by only 0.06%. When discharging at 9 A·g^−1^ current density, it can also show good rate performance, and the discharge capacity reaches 550 mAh g^−1^.

The researchers’ work on the functionalization of the separator layer on the electrostatic repulsion strategy enables the battery to exhibit an excellent inhibition of polysulfide ion shuttling, better electrolyte wettability, and significant improvement in battery electrochemical performance. All these studies fully demonstrated that this strategy is practical in alleviating the shuttle of polysulfides and has good application prospects.

### 4.4. Functionalized Separator with Spatial Barrier Effect

Studies using various strategies for the suppression of shuttle effects have shown that these different strategies have effectively mitigated polysulfide ion shuttles. Taking advantage of the difference in kinetic diameter between polysulfide ions and lithium ions, it is also an effective strategy to design the diaphragm channel that only allows lithium ions to pass through and to prevent polysulfide ions from passing through the diaphragm in size [75]. The research of this strategy is mainly based on the following: according to the large surface-to-diameter ratio characteristics of two-dimensional nanomaterials, the diameter of lithium-ion channels can be reduced by stacking materials, and the two-dimensional nanomaterials commonly used in lithium–sulfur batteries mainly include graphene [76,77], Mexene [78], and metal sulfides (such as WS_2_ [79,80] and MoS_2_ [81]).

Siyi Pan et al. (Figure 9) [82] synthesized a flower-like TiS_2_ nanostructure composed of ultrathin nanosheets and then recombined it with highly conductive carbon nanotubes (CNTs) to construct TiS_2_/CNT-modified separators, providing sufficient active sites and high-speed charge transmission channels for anchoring polysulfides to accelerate reaction kinetics. Through the charge and discharge test, the battery assembled by the TiS_2_/CNT-modified separator had an initial discharge specific capacity of 1012 mAh g^−1^ at a rate of 0.5 C, and the reversible capacity of the battery was still 848 mAh g^−1^ after 100 cycles. In addition, at a high sulfur load of 10.5 mg cm^−2^, the discharge specific capacity reached 878 mAh g^−1^. Cyclic voltammetry (CV) and electrochemical impedance spectroscopy (EIS) tests show that TiS_2_/CNT-modified separators can improve the mobility of lithium ions, accelerate reaction kinetics, and further reduce internal resistance and polarization voltage. The researchers also tried to prepare functionalized separators using two-dimensional structures of black phosphorus [83], montmorillonite [84], C_3_N_4_ [85], and other materials, which also made the battery show an excellent sulfur-fixation effect. A large number of articles have reported on the application of two-dimensional materials in the field of lithium–sulfur battery separators, which adequately show that the two-dimensional material modification of separators is a very effective strategy for improving the electrochemical properties of lithium–sulfur batteries.

### 4.5. Construction of Multifunctional New Separators

The conductive functionalization modification of the battery separator by various means and the functionalization of adsorption, catalytic activity, space steric resistance, etc., have been seen in a large number of studies, and these functionalization strategies can effectively alleviate the shuttle effect of polysulfides and are considered to significantly improve the electrochemical properties of lithium–sulfur batteries. However, the modification of existing commercial separators will inevitably increase the thickness of the separators and lead to a reduction in the battery energy density. The goal of researchers is to effectively inhibit the shuttle effect and improve the electrochemical performance on the premise of ensuring the energy density. Therefore, some researchers began to try to use functional organic polymers, organic metal skeleton materials (MOFs), and inorganic nanomaterials to design and construct new separators, hoping to alleviate the serious shuttle effect of lithium–sulfur batteries and the dendrite problem of the lithium metal anode and obtain more efficient, simple, and low-cost battery separators. In recent years, there have been a small number of reports on this work.

High-performance separators was prepared by using functional polymer as raw material. Li-Ling Chiu and Sheng-Heng Chung [86] synthesized a composite material using a gel polymer electrolyte and separator as a functional separator coated with a layer of poly (ethylene oxide) (PEO) and bis (trifluoromethanesulfonimide) lithium (LiTFSI). PEO/LiTFSI coated polypropylene film slows the diffusion of polysulfides and stabilizes liquid active materials in the cathode region of the battery while allowing the smooth transfer of lithium ions. Under the high-sulfur loading conditions of 2 mg cm^−2^, 4 mg cm^−2^, and 6 mg cm^−2^, the storage capacity of the lithium–sulfur battery is 1212 mAh g^−1^, 981 mAh g^−1^, and 637 mAh g^−1^, respectively, and the high reversible capacity of 534 mAh g^−1^ is maintained after 200 cycles, which proves that it can prevent the irreversible diffusion of polysulfides and improve the electrochemical stability and utilization rate of active substances.

Electrospinning technology is a mature preparation technology of non-woven fiber separator materials, which has played a very important role in the construction of one-dimensional nanostructured materials. Numerous studies have reported that this kind of technology is used for the preparation of lithium-ion battery separators [87], fuel cell separators, and other porous separator materials. Yin Hu et al. (Figure 10A) [88] prepared a nanofiber separator using electrospinning and chemically crosslinked poly(glycol)diacrylate grafted siloxane and polyacrylonitrile aqueous solution. After chemical crosslinking by siloxane, the prepared nanofiber separator has good mechanical properties and a high thermal stability, and the polar groups on the surface of the separator can interact with lithium ions, resulting in the uniform deposition of lithium and chemical absorption of polysulfide compounds. Compared with polypropylene (PP) separators, nanofiber separators can significantly improve the electrochemical cycling performance of lithium metal batteries and lithium batteries. To overcome the problem of polysulfide dissolution and the shuttle effect of lithium–sulfur batteries, Ayaulym Belgibayeva and Izumi Taniguchi(Figure 10B) [89] proposed to prepare an FS-SiO_2_/C-CNFM intermediate layer by electrospinning and heat treatment. The wide-pore physical adsorption of FS-SiO_2_/C-CNFM and chemical adsorption of strong polar interactions with Si-OH groups and N-doped carbon prevented the polysulfides from shuttling to the anode side. The results show that the insertion of the FS-SiO2/C-CNFM interlayer between the cathode and the separator improves the specific capacity, cycle life, and self-discharge behavior of lithium–sulfur batteries. At a high current density of 2 C, the initial discharge capacity of the FS-SiO_2_/C-cnfm mezzanine battery is 1030 mAh g^−1^, it remains at 664 mAh g^−1^ after 100 cycles, and the Coulomb efficiency is 100%. Han-Byeol Kim et al. [90] prepared vanadium nitride modified carbon nanofibers (VNCNFs) by electrospinning. The prepared VNCNFs are sandwiched between the cathode and the separator as an intermediate layer. Lithium–sulfur (Li-S) batteries with VNCNF sandwiches exhibit high initial charge–discharge capacities of 1452 and 1480 mAh g^−1^, corresponding to a Coulomb efficiency of more than 100% at 0.5 C magnification, and, after 400 cycles, the lithium–sulfur batteries still maintain a high capacity of 923 mAh g^−1^.

In addition, carbon materials are considered to be able to make up for the shortcomings of lithium batteries because of their light weight and strong conductivity. At present, researchers often use carbon materials, such as carbon nanotubes, carbon nanosheets, and graphene. However, the preparation process of these carbon materials is complicated and the cost is high, which is not suitable for mass production and application. Therefore, the majority of scientific research workers focus on the development of biomass carbon materials. The source of biomass carbon raw materials is rich, the preparation method is simple, and the price is low. The use of biomass carbon materials conforms to the policy of low-carbon environmental protection and sustainable development. From the perspective of structure, biochar materials have the advantages of a large specific surface area and high porosity, which can provide continuous electron transport channels and abundant active sites and play a key role in easing volume expansion. In terms of material properties, biochar materials have good electrical conductivity and chemical stability, which can solve the problem of material insulation. In terms of material composition, biomass materials are rich in amino acids, proteins, and so on. The proper control of the carbonization process will leave some polar heteroatomic groups in the finished product. Polar groups have a strong chemical adsorption capacity and can effectively adsorb polysulfide ions. Therefore, introducing biomass materials into separator applications can effectively inhibit some negative effects caused by the shuttle effect. The sources of biomass materials commonly used in batteries include dandelion [91], wheat flour [92,93], rice, etc. [94,95,96].

**Figure 10 polymers-15-00993-f010:**
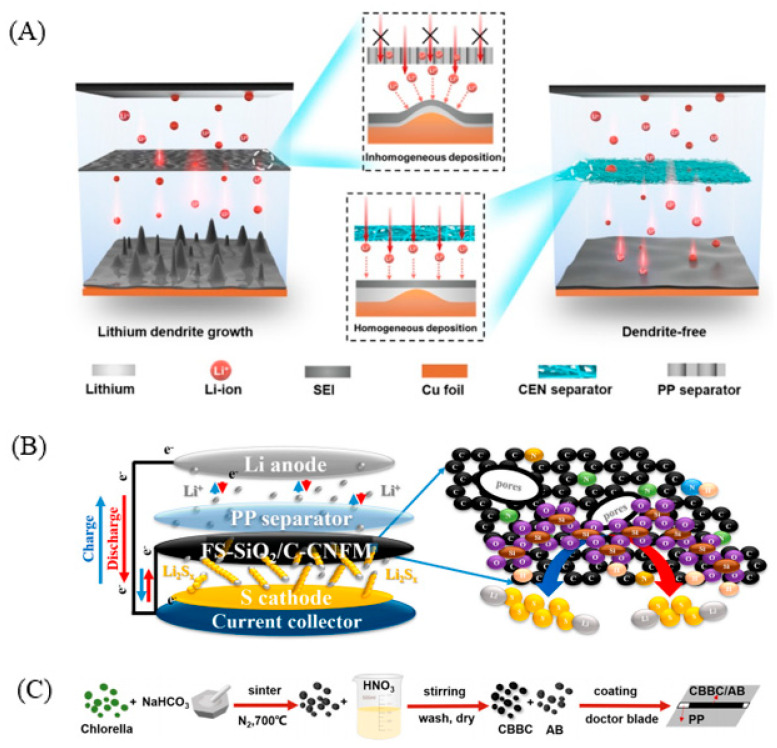
(**A**) Illustration of Li deposition on the Cu anode with the CEN and PP separators, and the homogenous Li-ion flux of the Li//CEN//Cu cells facilitates non-dendritic Li deposition. In contrast, the inhomogeneous Li-ion flux in uneven pores of the PP separators can result in dendritic growth and short circuit [88]. (**B**) Schematic illustration of the polysulfide dissolution, adsorption, and conversion by FS-SiO_2_/C-CNFM interlayer during the discharge–charge processes of lithium–sulfur cells [89]. (**C**) Schematic of the process of the preparation of CBBC-coated separator [97].

Qian Li et al. (Figure 10C) [97] adopted renewable Chlorella as the precursor of biomass-derived carbon. Firstly, the Chlorella was pretreated with NaHCO_3_ activation, and then the pretreated Chlorella was carbonized at high temperature. After carbonization, the heteroatoms were removed by pickling. Finally, N and O co-doped biochar was used to prepare the modified separator of regenerated Chlorella. The separator modified by Chlorella biochar played the role of a physical barrier and chemical adsorber, and the lithium polysulfide was successfully blocked in the cathode side. At the same time, some of the active substances were evenly dispersed on the surface of the separator to avoid the excessive accumulation of active substances. The separator modified by Chlorella biochar gives full play to the advantages of the high-capacity density of the lithium–sulfur battery and makes up for a series of problems caused by the shuttle effect. Using environment-friendly biomass carbon to modify the separator is one of the effective strategies to improve the electrochemical performance of lithium–sulfur batteries.

## 5. Summary and Expectation

As a new type of secondary battery system with great application prospect, the lithium–sulfur battery is favored by the majority of researchers. However, the commercial application of the lithium–sulfur battery faces many problems, such as the insulation of elemental sulfur and its short-chain sulfide, battery volume expansion during charge and discharge, repeated shuttle of polysulfide ions between anode and cathode, lithium metal anode corrosion and dendrite growth, etc. Among the many solution strategies, introducing a barrier layer of polysulfide shuttle from the separator layer between the cathode and the anode is considered an extremely effective research strategy. Based on the collation and summary of the research fruits in the field of the functionalization of separators over the years, this paper mainly divides the research into the following aspects: improving the electrochemical reaction activity and the utilization rate of the active substance by modifying the conductive layer of the separator; introducing adsorption materials to immobilize free polysulfides; introducing negatively charged groups to inhibit the passage of polysulfides through the separator by electrostatic repulsion; and introducing a catalytic function to accelerate the kinetic process of the conversion between polysulfides, optimize the separator negative contact interface, and improve the interface stability of the lithium anode. These research strategies have significant effects on mitigating the shuttle of polysulfides, improving the efficiency of active substance utilization, prolonging cycle life, and circulating stability and safety.

Although the research on multifunctional separators has made great progress, further research and commercial applications still have a broad research and development space. Judging from the current research fruits, the economical, efficient, environment-friendly, and safe high-performance lithium–sulfur battery separator is still an effective means to improve the performance of lithium–sulfur batteries and promote the practical application of lithium–sulfur batteries. From the perspective of the separator itself, it is still necessary to control the volume and mass load of the functional modification layer while obtaining an efficient sulfur suppression effect, to reduce the impact on energy density loss. From the perspective of the negative electrode, the functionalization of the separator needs to inhibit the growth of lithium dendrites in the negative electrode while ensuring the efficient inhibition of the shuttle of polysulfide ions, so as to ensure the safety and stability of the battery. From the perspective of the commercial application of separators, while taking into account the high performance of lithium–sulfur batteries, it is necessary to consider the production cost of functionalized separators from many aspects including low cost, high performance, and environmental friendliness. It can be found from the literature that the research in the field of lithium–sulfur battery separators is still dominated by low sulfur loading. Therefore, in order to obtain high-energy density lithium–sulfur batteries, further exploration can be carried out on high-sulfur batteries in the future, so as to meet practical application requirements. In addition, the research on functionalized separators in pouch batteries faces more practical problems. In the future, it is necessary to further investigate the performance of pouch batteries to provide more reliable solutions for practical and commercial applications of lithium–sulfur batteries.

## Figures and Tables

**Figure 1 polymers-15-00993-f001:**
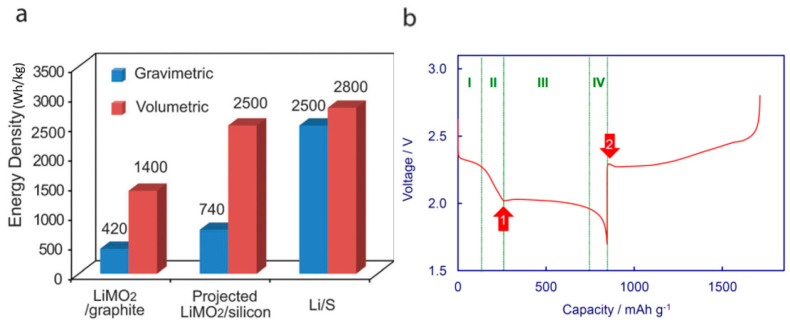
(**a**) Energy density [17] and (**b**) a typical discharge and charge voltage profile of Li/S cells [18].

**Figure 2 polymers-15-00993-f002:**
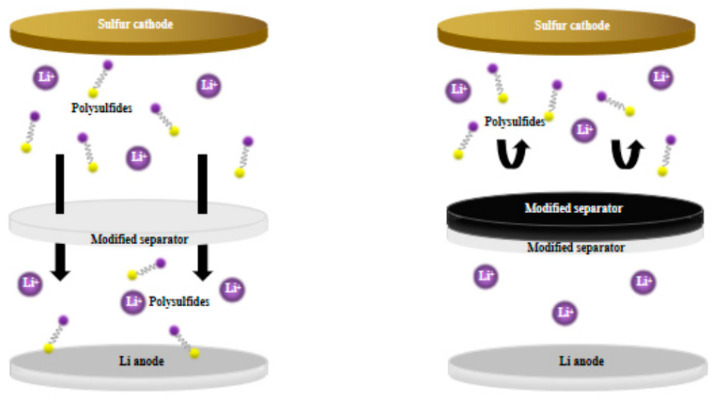
Schematic diagram of multifunctional separator.

**Figure 3 polymers-15-00993-f003:**
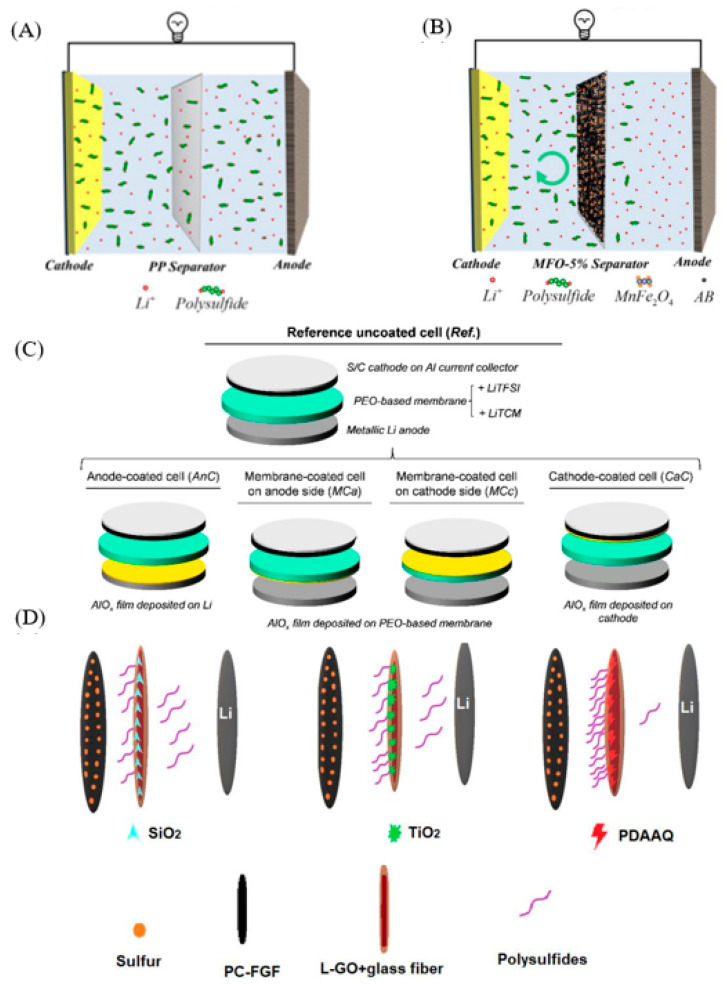
(**A**)The schematic of Li-S battery configured with pristine PP separator [45]. (**B**) The schematic of the Li-S battery configured with MFO-5% modified separator [45]. (**C**) Schematic representation of all battery configurations and coatings studied in this work [46]. (**D**) Schematic for the Li-S batteries configuration consisting of glass fiber separator with SiO_2_, TiO_2_, PDAAQ coating and L-GO binder [47].

**Figure 4 polymers-15-00993-f004:**
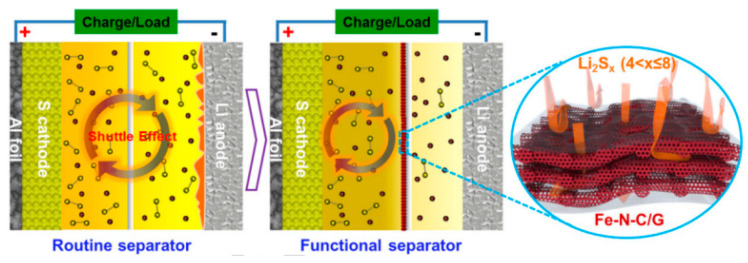
Schematic illustration of the Li-S batteries configured with a routine separator and functional separator [55].

**Figure 5 polymers-15-00993-f005:**
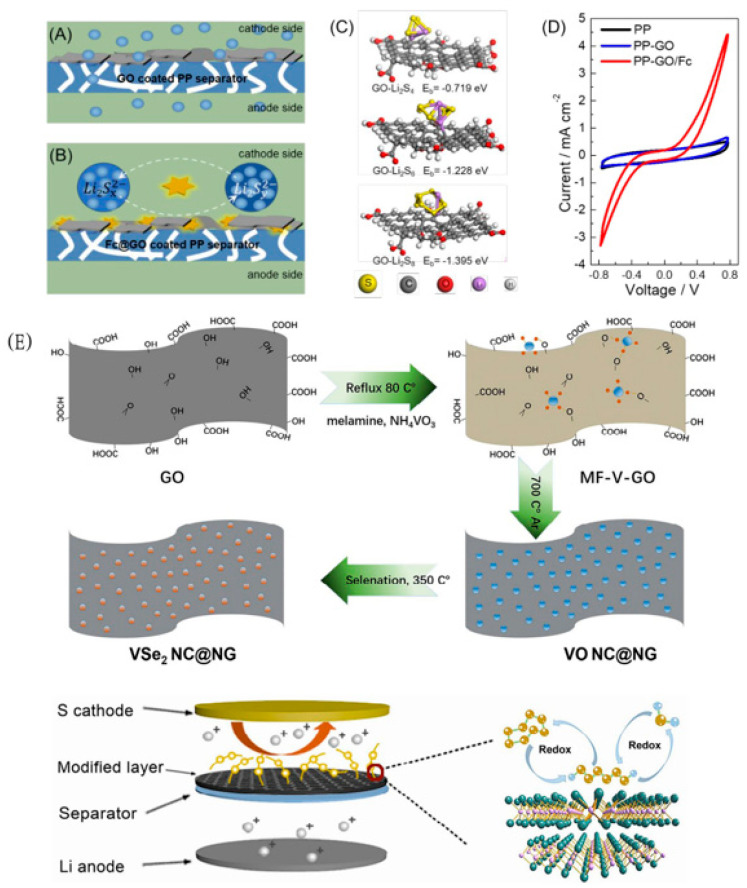
Schemes showing (**A**) unwanted polysulfide shuttling through the PP-GO separator and (**B**) suppressed polysulfide shuttling through the PP-GO separator due to the catalytic effect of Fc [59]. (**C**) Optimized configuration of polysulfide adsorption on model GO [59]. (**D**) CV curves of symmetric cells assembled with the PP, PP-GO, or PP-GO/Fc separator and Li_2_S_6_ containing electrolyte [59]. (**E**) Schematic illustration of the formation process of VSe_2_ NC@NG hybrid nanosheets (upper panel). The configuration of the LSBs with the VSe_2_ NC@NG modified separator as well as the illustration for the sulfur redox reaction on the surface of the VSe_2_ (lower panel) [60].

**Figure 6 polymers-15-00993-f006:**
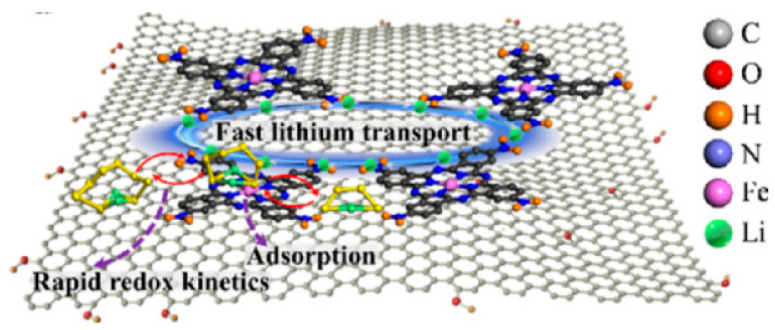
A schematic representation of the sulfur catalytic process on FeTaPc@rGO at the cathode/separator interface [69].

**Figure 7 polymers-15-00993-f007:**
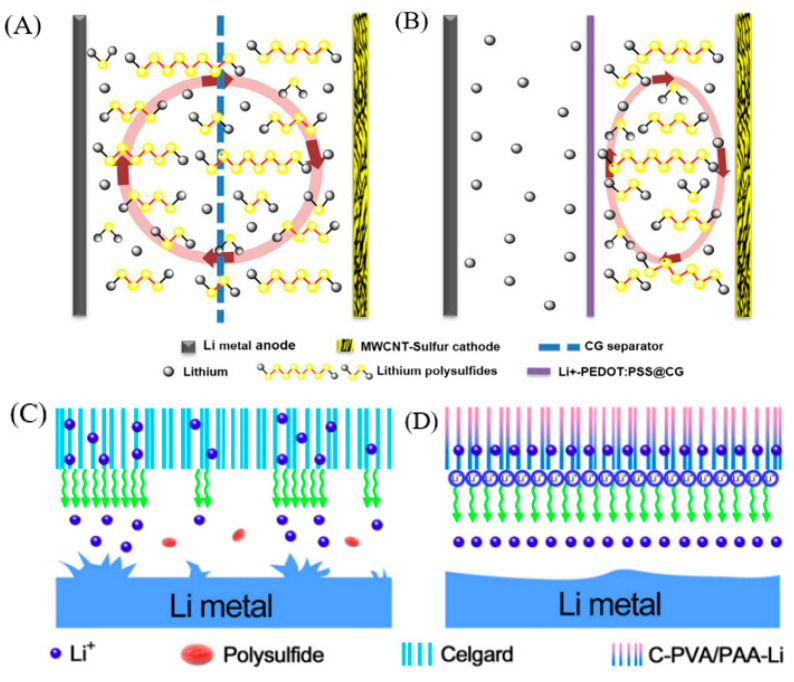
Schematic configuration of Li-S cells with (**A**) CG separator, in which polysulfides shuttle between the cathode and anode sides and (**B**) Li^+^—PEDOT:PSS@CG separator, in which the polysulfide anions are trapped within the cathode side [72]. (**C**) Schematic showing the formation of lithium dendrites due to slow and uneven Li^+^ deposition using Celgard separator [73]. (**D**) C-PVA/PAA-Li separator guides the rapid and uniform deposition of Li^+^ to avoid the formation of lithium dendrites [73].

**Figure 8 polymers-15-00993-f008:**
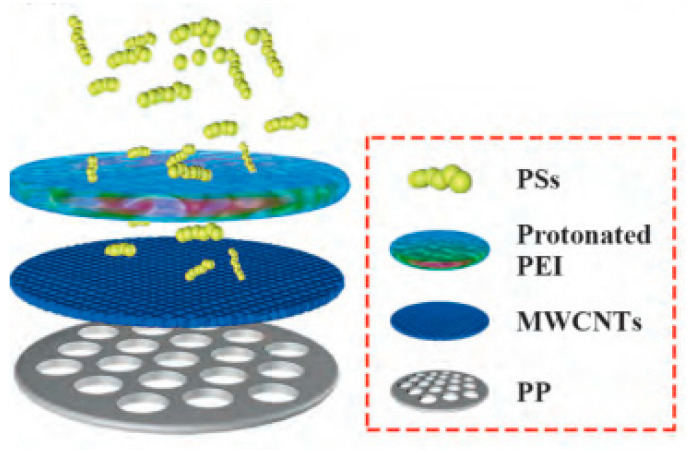
Schematic cell configuration of Li-S cell using the PMS separator on the polysulfides shuttling inhibition [74].

**Figure 9 polymers-15-00993-f009:**
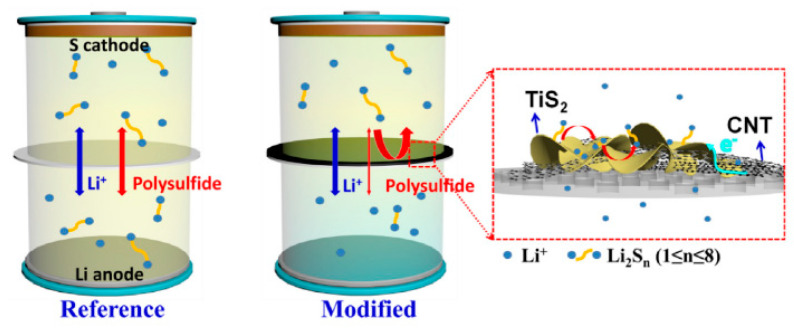
Schematic illustration of the polysulfide-trapping and electron-transporting mechanisms for the Li-S batteries employed by pristine separator and TiS_2_/CNT-modified separator [82].

## Data Availability

No new data were created or analyzed in this study. Data sharing is not applicable to this article.

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
