# Peer review of "Research Progress on Multifunctional Modified Separator for Lithium–Sulfur Batteries"

_polymers, 2023, doi:10.3390/polym15040993_

Round 1

Reviewer 1 Report

Summary:

This Review article polymers-2160466 titled, “Research Progress on Multifunctional Modified separator for Lithium-Sulfur Batteries,” reports the development, the functions, and the reach trends of the lithium-sulfur battery separator, especially the surface coating materials and technologies. The lithium-sulfur cells face the high cathode resistance and polysulfide diffusion. The reported multifunctional modified separator summarizes the important design of the separator modification to balance the high resistance of the sulfur cathode and the related high amount of diffusing polysulfides. Moreover, specific cases are reported in detail for the functional optimization of the polysulfide-trapping capability, fast electron transfer, and polysulfide conversion catalyst, etc. Almost 100 papers are highlighted and included, which inspire future researchers to explore this research area by having this review as a reference.

General comment:

The review article reports the development of the modified separator in the optimization of the lithium-sulfur battery cathode with stable electrochemistry. Sufficient reference papers are cited to build the fundamental understanding of the issues of lithium-sulfur cells and the solutions offered by the separator modification. Specific case study highlights the progresses and feasibility. Minor revisions are suggested to provide the necessary data and parameters. Hope the authors feel the comment useful.

Comments:

(1) The introduction of the reaction mechanism and the challenges of lithium-sulfur cells give a good summary of the issue and solution of lithium-sulfur batteries. Some revisions are suggested. In the equations (4) – (11), the states of materials are suggested to be written as the subscript letters of “(s)” and “(l),” the electron is “e-” in the writing, the “bS2 2-” is a typo, and the charges should be balanced. In Figure 1 (a), the energy density needs their unit in the plots. In Figure 1b, the discharge/charge voltage profile is incomplete one. A full reaction plot is suggested to support the discussion. For the “shuttle effect,” it is suggested to clarify the formation and reaction mechanism of polysulfide shuttle.

[Suggestion] Please revise the introduction sections mentioned above.

(2) Some case studies are highlighted and analyzed with the necessary parameters and performance. This gives the better understandings of the development. Thus, it is suggested to report the important information that is related to the separator modification and shows the effect. First, the sulfur loading and content of the cathode are suggested to be reported if the original papers have mentioned those necessary data. Second, the thickness and mass loading of the coating materials are suggested to be summarized since they are critical experimental data.

[Suggestion] Please report the suggested information if it is possible to collect from the original papers.

(3) The highlighted cases in the figures and the main text give the configuration and functions as well as the effects and the resulting electrochemical performances. Since this review article collects the papers used different cathode and cell fabrication parameters, it is suggested to give a normalization summary. For example, the areal capacity and gravimetric discharge capacity are suggested. The energy density values are also important. However, in consideration of the original paper might mostly report the basic active-material capacity. This comment is suggested for the authors.

[Suggestion] Please report the suggested important cell performance data for a better understanding if it is possible.

(4) In general, the manuscript is well organized. One suggestion is to rewrite the sentences and paragraphs that are similar to a recently-reported paper “Effects of catalysis and separator functionalization on high energy lithium sulfur batteries: A complete review.” The published review paper is therefore suggested to be cited as a strong support reference.

[Suggestion] Please revise the manuscript by considering the suggested writing modification.

(5) In all, the engineering design and function of the modified separator in the lithium-sulfur cells are necessary for the discussion and the introduction. To support the discussion, some pioneer and recent works are suggested to show the research trends and progresses: (Advanced Functional Materials, 2014, 24, 5299-5306; The Journal of Physical Chemistry Letters, 2014, 5, 1978-1983; ACS Nano, 2015, 9, 3002-3011: initial works in modified lithium-sulfur battery separator; Acta Chimica Sinica, 2017, 75, 173-188; Sustainable Energy Fuels, 2021, 5, 5656-5671; Membranes 2022, 12, 790; Molecules 2022, 27, 228: summary of coating materials and functions of the modified separators)

[Suggestion] Please discuss the engineering design and material development of the modified separators in the introduction with the support of the suggested references.

Reviewer 2 Report

1. There are many grammatical and typing mistakes in the manuscript.

2. The terminology should be unified;

polylithium sulfide should be changed to lithium polysulfide.

2. There is no explanation of Fig 1 a.

3. Fig 1b should be divided into 4 regions as shown in the manuscript.

4. in section 4.3, lithium-rated should be changed to lithiated,  PDA-Li should be PAA-Li,

5. Some of the abbreviation requires full name. what is PMS??

6. In section 4.5 second paragraph, no number in loading mass.

6. In page 15, ref 93 is related with N-doped carbon.
